# Distribution of pediatric intensive care beds in Brazil: Regional inequalities and implications for equity in the public health system

Luís Felipe Ribeiro Soares[1,2]*, Maria Auxiliadora Souza Mendes Gomes[3], Saint Clair Gomes Júnior[3]

**1** Paulo Niemeyer State Institute of the Brain (IECPN), Department of Pediatrics, Rio de Janeiro/RJ, Brazil, **2** Martagão Gesteira Institute of Childcare and Pediatrics (IPPMG) – Federal University of Rio de Janeiro (UFRJ) , Rio de Janeiro/RJ, Brazil, **3** Fernandes Figueira National Institute of Women, Children and Adolescents' Health (IFF/Fiocruz), Rio de Janeiro/RJ, Brazil

* lufesoares@hotmail.com

## Abstract

### Background

Pediatric Intensive Care Units (PICUs) play a critical role in reducing child mortality, but their distribution across Brazil remains markedly uneven. Understanding how the spatial and technological distribution of PICU beds relates to regional child health indicators is essential for promoting equity within the Unified Health System (SUS).

### Objective

To examine the national distribution of PICU beds in Brazil, identify regional inequality patterns, and explore their association with pediatric mortality to inform equitable health planning.

### Methods

This cross-sectional ecological study analyzed data from the *National Register of Health Establishments (CNES)* and demographic indicators from the *Brazilian Institute of Geography and Statistics (IBGE)*, encompassing all 5,570 municipalities and 450 health regions as of July 2023. Variables included the number and type of PICU beds (types I, II, and III), resident population, proportion of individuals under 14 years, and mortality in this age group. A multivariate cluster analysis was applied to group health regions by similarities in bed availability and mortality profiles, using Bayesian and Akaike Information Criteria and the Silhouette coefficient to define cluster quality.

### Results

A total of 6,489 PICU beds were identified in 708 healthcare providers, with 50% allocated to SUS. Most beds were type II (64.7%), followed by type III (18.6%) and type

**Data availability statement:** All raw data required to replicate the results of this study (including the values used in tables, statistical analyses, and figures) will be uploaded as Supporting Information files. These data were obtained from publicly available sources (DATASUS and IBGE), and the processed datasets used in the analyses are being made fully accessible to ensure transparency and reproducibility.

**Funding:** The author(s) received no specific funding for this work.

**Competing interests:** The authors have declared that no competing interests exist.

I (16.6%). Six clusters were identified, revealing marked inequities: regions with higher proportions of type III beds had lower child mortality (16.5 per 100,000), while areas with predominantly type I beds or none had higher mortality (up to 25.0 per 100,000). Notably, 259 health regions (46%) lacked any PICU beds, affecting over 53 million residents.

## Conclusions

The unequal distribution of PICU beds across Brazil reflects structural inequities in access to pediatric critical care. Policies should prioritize regionalized, quality-based allocation, intergovernmental coordination, and integration of health data systems to guide equitable expansion of pediatric intensive care capacity within the SUS.

---

## Introduction

Pediatric intensive care refers to a set of technologies and procedures delivered by highly specialized healthcare professionals, aimed at reducing mortality and morbidity among critically ill children and adolescents [1]. The effectiveness of these interventions, however, relies not only on clinical expertise but also on the proper organization of Pediatric Intensive Care Units (PICUs). To ensure optimal outcomes, PICUs must meet minimum standards of quality and technological capacity, and be strategically located and sufficiently available to address local healthcare needs [1–3].

In Brazil, the Brazilian Health Regulatory Agency (Anvisa) has defined minimum operational standards for all Intensive Care Units (ICUs), including pediatric units, to ensure safe and effective care delivery [4]. These units are designated as "critical areas intended for the hospitalization of seriously ill patients who require continuous specialized professional attention, specific materials, and technologies necessary for diagnosis, monitoring, and therapy." Pediatric Intensive Care Units (PICUs) therefore play a crucial role in guaranteeing the right to health, as established by the Brazilian Federal Constitution of 1988 and Law No. 8,080 of September 19, 1990, which established the Unified Health System (SUS). Their importance is further highlighted in the context of Brazil's epidemiological transition, which requires greater access to specialized pediatric care [5–8].

Specifically for PICUs, their characterization and operation have been regulated by specific laws and normative acts since 2000. These regulations stipulate that the calculation of the number of PICU beds must consider technical aspects such as: scientific evidence; clinical and therapeutic protocols related to care lines based on literature reviews; analysis of the structure and performance of the installed capacity of national hospitals; comparison between the actual situation of hospitals and ideal, recognized standards; parameter adjustments to be used in bed estimation, and the application of simulation models [9–10].

Despite regulations, access to pediatric intensive care in Brazil is difficult and complex. The national dimension of the country, socioeconomic and regional differences in health infrastructure contribute to an unequal distribution of services [2,11–14].

Locations with lower technological density often have difficulty providing highly complex care, resulting in patient transfers that can increase the risk of morbidity and mortality [15]. This study describes the national distribution and complexity of Pediatric Intensive Care Unit (PICU) beds in Brazil and examines their relationship with child mortality. Data were obtained from different official government sources. These databases together offer a comprehensive view of the Brazilian health system.

## Materials and methods

### Design, population, and location

Data for this ecological study were obtained from official, open-access databases managed by the Brazilian Ministry of Health and the Brazilian Institute of Geography and Statistics (IBGE). This is a cross-sectional study based on data recorded in the National Register of Health Establishments (Cadastro Nacional de Estabelecimentos de Saúde – CNES) by healthcare providers from all Brazilian states. The CNES, available through the DATASUS platform, compiles detailed information on infrastructure, human resources, and installed capacity for all public and private healthcare facilities nation-wide. The present analysis considered data referring to the number of Pediatric Intensive Care Unit (PICU) beds of types I, II, and III, as of July 2023, selected because this month coincides with the seasonal peak of pediatric hospitalizations in Brazil. These records are stored in the LTUFYYMM table, where *LT* refers to beds, *UF* to the state of Brasil, and YY*MM* to the reference year and month, identified by the codes 77 (PICU I), 78 (PICU II), and 79 (PICU III) in the *CODLEITO* field (Number of Supplementary Beds – SUS).

Mortality data were obtained from the Mortality Information System (Sistema de Informações sobre Mortalidade – SIM), which compiles all officially registered deaths in Brazil, including information on age, cause, and place of residence. For this study, mortality records corresponding to the year 2023 were analyzed, enabling the calculation of child and age-specific mortality indicators. Demographic data were extracted from the Brazilian Institute of Geography and Statistics (IBGE), which provides official population estimates and age-structure data by municipality. These demographic data served as denominators for computing age-specific mortality rates among individuals under 14 years of age.

Together, these sources provide a comprehensive and routinely updated dataset for monitoring healthcare infrastructure and health outcomes across Brazil. The authors did not have access to any information that could identify individual participants during or after data collection.

### Context

The CNES data are published monthly by DATASUS and may vary over the year according to information provided by healthcare providers. For this reason, this study uses July 2023 as the reference month, assuming that decisions to increase PICU bed capacity are more likely during this period due to the higher prevalence of Severe Acute Respiratory Syndrome (SARS) cases in autumn and winter months. To ensure robustness, a sensitivity analysis was performed using data from all months of 2023, showing a variation of only 0.83% in the total number of PICU beds and 0.31% in beds belonging to the SUS network throughout the year. For this study, only the tables related to healthcare provider identification and the number of beds offered (types I, II, and III) were used and linked via the provider identification code present in both datasets.

The categorization of PICU beds into levels of complexity (I, II, and III) is defined by regulations of the Unified Health System (SUS) and is based on technological density, the level of care complexity, and the resources needed to treat critically ill pediatric patients [4,10]. This classification defines three levels of technological and structural complexity—Type I, Type II, and Type III—which correspond approximately to the Level I, II, and III PICU framework recommended by the American Academy of Pediatrics (AAP) for the organization of pediatric critical care systems.

• Type I PICU: These units provide basic intensive care for children with less severe conditions who require continuous monitoring and short-term ventilatory or hemodynamic support. They are typically located in smaller hospitals or regional

facilities, often in areas where access to tertiary care centers is limited. Their infrastructure supports basic cardiorespiratory monitoring, pulse oximetry, and noninvasive ventilation, with care delivered by pediatricians trained in intensive support.

- Type II PICU: These are intermediate-complexity units capable of managing children with more severe or unstable clinical conditions who require continuous invasive monitoring and prolonged mechanical ventilation. These units generally operate in medium- or large-sized hospitals with a multidisciplinary team that includes pediatric intensivists, nurses with critical care training, and access to diagnostic imaging and laboratory facilities.

- Type III PICU: Representing high-complexity tertiary or quaternary units, Type III facilities care for critically ill children with life-threatening or multisystem conditions requiring advanced life-support technologies. These include invasive hemodynamic monitoring, renal replacement therapy, extracorporeal membrane oxygenation (ECMO), and advanced ventilatory support. Type III units are typically located in major referral hospitals, teaching hospitals, or specialized pediatric centers, staffed by board-certified intensivists and subspecialty consultants, with full access to surgical, imaging, and laboratory services 24/7.

This tiered classification ensures a hierarchical organization of pediatric critical care, linking lower- and higher-complexity units through regionalized referral networks—an approach consistent with international standards recommended by the AAP and the World Federation of Pediatric Intensive and Critical Care Societies (WFPICCS).

### Eligibility criteria

All records from public and private hospital units registered in CNES for the reference month/year 06/2023 were included. Among these records, only those marked with codes 77, 78, and 79 (corresponding to PICU types I, II, and III) in the CODLEITO field were considered.

### Variables

Analyses were performed using the following variables: municipality code, federal unit (state), health region, resident population in the municipality, proportion of individuals under 14 years old, mortality rate of those under 14, and the proportion of PICU beds by complexity level (I, II, or III) in the municipality. Information on the federal unit and health region was retrieved from geographic information databases provided by DATASUS by linking the municipality code to corresponding state and region codes. The proportion of the population under 14 years old was calculated by dividing the number of residents under 14 by the total municipal population (data from IBGE). The mortality rate for individuals under 14 was calculated by dividing the total number of deaths under 14 (from the Mortality Information System – SIM) by the total population in the municipality (from IBGE).

### Database construction

The database was built using public CNES files provided by DATASUS (https://datasus.saude.gov.br/cnes-estabelecimentos). These files are in.dbc format and organized by state and reference date (year/month). The TabWin software (also from DATASUS) allows conversion from.dbc to.dbf, which can then be saved in.csv or.xlm format to be read by various database software. For this study, Microsoft® Power BI's native PowerQuery tool was used to merge files from different states, link CNES data with population and municipal data, and create the final table with data aggregated by health region.

### Data analysis

Categorical data were evaluated based on absolute and relative frequency of the considered variables.

A multivariate hierarchical cluster analysis was used to identify similarity patterns among the evaluated data. The following variables were considered: number of municipalities and health regions, total population and percentage of residents under 14, general mortality and mortality rate of those under 14, number of healthcare providers with available PICU beds, total number of PICU beds, and number of beds by complexity level. The variables used in the clustering process were standardized using the mean of the percentage distribution of type I, II, and III beds. The number of clusters was defined based on model fit and internal validity metrics, including the Bayesian Information Criterion (BIC) and the Silhouette coefficient [16,17]. The latter evaluates cohesion and separation among formed groups, ranging from −1 to +1, where values above 0.5 indicate good partitioning and values below 0.2 suggest weak cluster structure.

## Ethical considerations

This study used exclusively secondary data from public databases—DATASUS (National Health Database) and IBGE (Brazilian Institute of Geography and Statistics)—which are freely accessible and fully anonymized prior to public release. No individual-level or personally identifiable information was accessed or used at any point in the research process.

The research protocol was submitted to the **Research Ethics Committee of the Instituto Fernandes Figueira, Fundação Oswaldo Cruz (IFF/Fiocruz)**, which reviewed the study and confirmed that it is **exempt from formal ethical approval** in accordance with Brazilian regulations for research using anonymized, publicly available data.

In accordance with Resolution No. 510 of April 7, 2016, issued by the National Health Council (CNS) of the Brazilian Ministry of Health, studies that use publicly available and anonymized data are exempt from ethical review and do not require informed consent. Therefore, the requirement for participant consent was waived by regulation, and no consent—verbal or written—was required or obtained [18].

The study did not involve direct interaction with human participants, and no minors or vulnerable populations were involved. All analyses were conducted with aggregated data at the municipal and regional levels, ensuring full protection of privacy and confidentiality.

## Results

The analysis considered 5,570 municipalities across 450 health regions, encompassing a total population of 173,143,050 people. Of these, 19.8% were under 14 years old, with a mortality rate of 22.5 per 100,000 inhabitants during the study period. According to CNES data for July 2023, there were 708 service providers in the country (Table 1). These providers registered a total of 6,489 PICU beds, proportionally distributed by complexity level as follows: type I (16.6%), type II (64.7%), and type III (18.6%). Of these, 3,200 beds (approximately 50%) were indicated as being part of the SUS and were spread across 133 health regions in 303 municipalities. States such as São Paulo and Minas Gerais in the Southeast had the highest number of beds (both SUS and non-SUS), while smaller states like Roraima and Amapá had the lowest availability.

Table 2 presents the distribution of PICU beds in Brazil by region, hospital typology (PICU I, II, and III), and management type (SUS and non-SUS). The regional breakdown is as follows:

- **North Region** – 493 PICU beds: 8.5% type I, 89.9% type II, and 1.6% type III. The region has a population of 17,354,884, with 25.2% under 14 years old. The mortality rate for this age group is 36.6 per 100,000 inhabitants.

- **Northeast Region** – 1,329 beds: 9.0% type I, 82.1% type II, and 9.0% type III. The population is 54,658,515, of which 21.1% are under 14. The under-14 mortality rate is 25.8.

- **Southeast Region** – 3,270 beds: 22.0% type I, 53.9% type II, and 24.0% type III. This region has 84,840,113 residents, with 18.0% under 14. The mortality rate for children under 14 is 18.8 per 100,000.

- **South Region** – 748 beds: 3.9% type I, 63.8% type II, and 32.4% type III. Total population is 29,937,706, with 18.5% under 14. Mortality rate for this group is 17.4 per 100,000.

**Table 1. Distribution of PICU beds in Brazil, according to health regions, hospital typology, and sociodemographic and mortality characteristics.**

| Region/State | Health regions[1] | Municipalities[2] | Population[3] | % < 14 years* | Deaths** | Deaths < 14 years§ | Providers# | PICU Beds£ | PICU I¢ | PICU II¢¢ | PICU III¢¢¢ |
|---|---|---|---|---|---|---|---|---|---|---|---|
| **Brasil** | **450** | **5.570** | **203.080.756** | **19,8** | **87.306** | **22,5** | **708** | **6.489** | **16,6** | **64,7** | **18,6** |
| **North** | **45** | **450** | **17.354.884** | **25,2** | **6.346** | **36,6** | **53** | **493** | **8,5** | **89,9** | **1,6** |
| AC | 3 | 22 | 830.018 | 26,6 | 346 | 41,7 | 3 | 23 | 0,0 | 100,0 | 0,0 |
| AM | 9 | 62 | 3.941.613 | 27,3 | 1.640 | 41,6 | 12 | 119 | 4,2 | 91,6 | 4,2 |
| AP | 3 | 16 | 733.759 | 27,0 | 340 | 46,3 | 1 | 38 | 0,0 | 100,0 | 0,0 |
| PA | 13 | 144 | 8.120.131 | 24,5 | 2.744 | 33,8 | 23 | 191 | 13,1 | 85,3 | 1,6 |
| RO | 7 | 52 | 1.581.196 | 22,0 | 479 | 30,3 | 8 | 64 | 18,8 | 81,3 | 0,0 |
| RR | 2 | 15 | 636.707 | 29,2 | 384 | 60,3 | 1 | 15 | 0,0 | 100,0 | 0,0 |
| TO | 8 | 139 | 1.511.460 | 23,2 | 413 | 27,3 | 5 | 43 | 0,0 | 100,0 | 0,0 |
| **Northeast** | **133** | **1.794** | **54.658.515** | **21,1** | **14.093** | **25,8** | **139** | **1.329** | **9,0** | **82,1** | **9,0** |
| AL | 10 | 102 | 3.127.683 | 22,8 | 849 | 27,1 | 12 | 92 | 5,4 | 80,4 | 14,1 |
| BA | 28 | 417 | 14.141.626 | 20,2 | 3.699 | 26,2 | 25 | 258 | 1,9 | 88,8 | 9,3 |
| CE | 22 | 184 | 8.794.957 | 20,5 | 1.888 | 21,5 | 19 | 178 | 7,3 | 91,0 | 1,7 |
| MA | 19 | 217 | 6.776.699 | 24,3 | 2.135 | 31,5 | 13 | 111 | 18,0 | 64,0 | 18,0 |
| PB | 16 | 223 | 3.974.687 | 20,8 | 1.061 | 26,7 | 20 | 139 | 18,0 | 64,7 | 17,3 |
| PE | 12 | 185 | 9.058.931 | 20,9 | 2.248 | 24,8 | 27 | 378 | 5,6 | 88,4 | 6,1 |
| PI | 11 | 224 | 3.271.199 | 20,8 | 886 | 27,1 | 9 | 66 | 22,7 | 63,6 | 13,6 |
| RN | 8 | 167 | 3.302.729 | 19,8 | 665 | 20,1 | 9 | 76 | 17,1 | 82,9 | 0,0 |
| SE | 7 | 75 | 2.210.004 | 21,2 | 662 | 30,0 | 5 | 31 | 6,5 | 83,9 | 9,7 |
| **Southeast** | **164** | **1.668** | **84.840.113** | **18,0** | **15.957** | **18,8** | **375** | **3.270** | **22,0** | **53,9** | **24,0** |
| ES | 3 | 78 | 3.833.712 | 19,3 | 853 | 22,2 | 16 | 118 | 16,1 | 83,9 | 0,0 |
| MG | 89 | 853 | 20.539.989 | 18,1 | 3.714 | 18,1 | 63 | 481 | 15,4 | 73,2 | 11,4 |
| RJ | 9 | 92 | 16.055.174 | 17,8 | 3.340 | 20,8 | 84 | 814 | 23,8 | 54,7 | 21,5 |
| SP | 63 | 645 | 44.411.238 | 18,0 | 8.050 | 18,1 | 212 | 1.857 | 23,4 | 46,7 | 29,9 |
| **South** | **69** | **1.191** | **29.937.706** | **18,5** | **5.212** | **17,4** | **85** | **748** | **3,9** | **63,8** | **32,4** |
| PR | 22 | 399 | 11.444.380 | 19,2 | 2.090 | 18,3 | 39 | 276 | 3,3 | 55,1 | 41,7 |
| RS | 30 | 497 | 10.882.965 | 17,5 | 1.788 | 16,4 | 26 | 264 | 2,3 | 58,3 | 39,4 |
| SC | 17 | 295 | 7.610.361 | 18,7 | 1.334 | 17,5 | 20 | 208 | 6,7 | 82,2 | 11,1 |
| **Central West** | **39** | **467** | **16.289.538** | **20,9** | **4.090** | **25,1** | **56** | **649** | **26,0** | **65,6** | **8,3** |
| DF | 1 | 1 | 2.817.381 | 19,0 | 518 | 18,4 | 14 | 252 | 51,6 | 35,7 | 12,7 |
| GO | 18 | 246 | 7.056.495 | 20,3 | 1.607 | 22,8 | 18 | 190 | 16,3 | 83,7 | 0,0 |
| MS | 4 | 79 | 2.757.013 | 22,0 | 757 | 27,5 | 6 | 57 | 0,0 | 75,4 | 24,6 |
| MT | 16 | 141 | 3.658.649 | 22,7 | 1.208 | 33,0 | 18 | 150 | 5,3 | 89,3 | 5,3 |

[1]Health regions, determined by the Ministry of Health, scope of ambulatory and hospital high complexity care

[2]Number of municipalities per Federation Unit and health region, according to 2022 Brazilian Institute of Geography and Statistics (IBGE) data

[3]Total resident population by municipalities referenced in health regions, according to 2022 IBGE data

*Percentage of the population under 14 years old, according to 2022 IBGE data

**Total number of deaths by municipality and health region, according to the Ministry of Health Mortality Information System (SIM)

§Death rate under 14 years per 100,000 inhabitants, according to SIM

#Number of institutions with PICU beds available to SUS (Brazil's public health system) as listed in the National Health Establishments Registry (CNES)

£Pediatric Intensive Care Unit (PICU) beds listed in CNES

¢Percentage of type I PICU beds listed in CNES

¢¢Percentage of type II PICU beds listed in CNES

¢¢¢Percentage of type III PICU beds listed in CNES

**Table 2. Distribution of PICU beds in Brazil, according to regions, hospital typology, and management.**

| Reion | PICU I | | PICU II | | PICU III | |
|---|---|---|---|---|---|---|
| | SUS | Non-SUS | SUS | Non-SUS | SUS | Non-SUS |
| North | 0 | 42 | 308 | 135 | 1 | 7 |
| Northeast | 20 | 99 | 691 | 400 | 65 | 54 |
| Central West | 8 | 161 | 251 | 175 | 21 | 33 |
| Southeast | 40 | 681 | 955 | 809 | 361 | 424 |
| South | 6 | 23 | 318 | 159 | 155 | 87 |
| **Total** | **74** | **1.006** | **2.523** | **1.678** | **603** | **605** |

- **Central-West Region** – 649 beds: 26.0% type I, 65.6% type II, and 8.3% type III. The population is 16,289,538, with 20.9% under 14 years. Mortality rate for this group is 25.1 per 100,000.

A multivariate analysis based on the number of pediatric intensive care beds and the mortality of children under 14 identified six groups (clusters), revealing important regional patterns and disparities (Table 3). These groups were defined based on the proportionality of registered PICU beds in each health region and observed child mortality rates. They are described as follows:

- **Cluster 1**: Highest number of PICU beds (2,676) across 334 providers, mainly in the Southeast and Northeast. Predominance of type II beds (92.3%). Mortality rate under 14: 23.9 per 100,000.

- **Cluster 2**: 48 PICU beds from five providers, mostly type I (85.4%). Mortality rate under 14: 21.1 per 100,000.

- **Cluster 3**: 1,329 beds in 20 health regions, with 58% type II and 33.6% type I. Providers concentrated in São Paulo and the Federal District (53 and 14 of the 135 providers). Mortality rate under 14: 20.4 per 100,000.

- **Cluster 4**: Second-largest number of beds (2,160), with 35.8% type III. States of Rio de Janeiro and São Paulo host 164 of the 204 health providers in this group. Mortality rate under 14: 19.1 per 100,000.

- **Cluster 5**: 276 beds in 153 grouped health regions. High frequency of type III beds (76.4%), mostly in the South and Southeast. Lowest mortality rate under 14: 16.5 per 100,000.

- **Cluster 6**: Comprises health regions without any registered PICU beds. Mortality rate under 14: 25.0 per 100,000.

The spatial distribution of pediatric intensive care resources across Brazil is illustrated in Fig 1. The map on the left displays the absolute number of PICU beds per Health Region in 2023, evidencing a marked territorial imbalance, with large areas—especially in the North, Northeast, and parts of the Midwest—holding few or no beds, while higher concentrations are found in the Southeast and Southern regions. The map on the right presents the regional classification derived from the clustering analysis, which groups Health Regions according to their PICU availability and service characteristics. Regions classified in Cluster 1 occupy the largest national territory, whereas higher-performing clusters are restricted to a small number of Health Regions, primarily in more developed areas of the country.

## Discussion

The results of this study achieved the intended objective and corroborate other studies that have already demonstrated an unequal distribution of PICU beds across the evaluated health regions [12–15]. However, this study advanced the analysis through a multivariate approach that allowed for a deeper exploration of inequality patterns, revealing, for example, regions with a higher or lower proportion of PICU beds by typology. Moreover, although causal relationships cannot be inferred, the under-14 mortality rate observed in each cluster appears to be correlated in some way with the distribution of these beds.

**Table 3. Distribution of PICU beds in Brazil, according to health regions, hospital typology, and sociodemographic and mortality characteristics (multivariate analysis).**

| Region/State | Health region¹ | Municipalities² | Population³ | %<14 years* | Deaths<14 years§ | Providers# | PICU Beds£ | PICU I¢ | PICU II¢¢ | PICU III¢¢¢ |
|---|---|---|---|---|---|---|---|---|---|---|
| **Cluster 1** | **140** | **2.037** | **75.391.806** | **20,2** | **23,9** | **334** | **2.676** | **3,8** | **92,3** | **3,9** |
| North | 19 | 205 | 9.965.738 | 25,6 | 38,4 | 40 | 364 | 2,5 | 96,2 | 1,4 |
| Northeast | 29 | 450 | 24.841.652 | 20,2 | 24,5 | 110 | 1.090 | 4,8 | 88,5 | 6,7 |
| Southeast | 48 | 590 | 20.540.823 | 18,3 | 19,0 | 102 | 563 | 3,2 | 95,0 | 1,8 |
| South | 33 | 615 | 13.263.308 | 18,6 | 18,1 | 50 | 346 | 0,6 | 96,8 | 2,6 |
| Central West | 11 | 177 | 6.780.285 | 20,9 | 25,9 | 32 | 313 | 6,4 | 91,1 | 2,6 |
| **Cluster 2** | **4** | **93** | **1.534.401** | **19,4** | **21,1** | **5** | **48** | **85,4** | **0,0** | **14,6** |
| North | 1 | 14 | 324.844 | 20,9 | 23,4 | 1 | 10 | 100,0 | 0,0 | 0,0 |
| Northeast | 1 | 42 | 365.614 | 19,7 | 15,3 | 2 | 22 | 68,2 | 0,0 | 31,8 |
| Southeast | 1 | 19 | 405.916 | 17,2 | 19,0 | 1 | 6 | 100,0 | 0,0 | 0,0 |
| Central West | 1 | 18 | 438.027 | 20,3 | 26,0 | 1 | 10 | 100,0 | 0,0 | 0,0 |
| **Cluster 3** | **20** | **199** | **26.440.187** | **18,7** | **20,4** | **135** | **1.329** | **33,6** | **58,0** | **8,4** |
| North | 1 | 5 | 1.978.620 | 19,0 | 24,4 | 12 | 119 | 19,3 | 78,2 | 2,5 |
| Northeast | 3 | 27 | 2.349.722 | 20,6 | 24,9 | 18 | 127 | 25,2 | 66,9 | 7,9 |
| Southeast | 13 | 146 | 18.501.745 | 18,3 | 19,3 | 86 | 794 | 31,7 | 59,8 | 8,4 |
| Central West | 3 | 21 | 3.610.100 | 19,4 | 21,0 | 19 | 289 | 48,1 | 40,8 | 11,1 |
| **Cluster 4** | **16** | **210** | **39.673.514** | **17,9** | **19,1** | **204** | **2.160** | **22,0** | **42,2** | **35,8** |
| Northeast | 1 | 5 | 1.477.303 | 20,4 | 31,7 | 8 | 81 | 24,7 | 50,6 | 24,7 |
| Southeast | 10 | 102 | 30.564.136 | 17,5 | 18,8 | 173 | 1.806 | 24,4 | 40,6 | 35,0 |
| Central West | 1 | 34 | 1.496.423 | 21,5 | 23,4 | 4 | 37 | 0,0 | 62,2 | 37,8 |
| **Cluster 5** | **11** | **135** | **6.866.344** | **17,8** | **16,5** | **29** | **276** | **5,8** | **17,8** | **76,4** |
| Northeast | 1 | 8 | 186.414 | 23,8 | 20,9 | 1 | 9 | 0,0 | 0,0 | 100,0 |
| Southeast | 6 | 64 | 2.443.048 | 17,6 | 16,6 | 13 | 101 | 5,0 | 19,8 | 75,2 |
| South | 4 | 63 | 4.236.882 | 17,7 | 16,2 | 15 | 166 | 6,6 | 17,5 | 75,9 |
| **Cluster 6** | **259** | **2.896** | **53.174.504** | **21,3** | **25,0** | **1** | **0** | **–** | **–** | **–** |
| North | 24 | 226 | 5.085.682 | 27,0 | 38,5 | 0 | 0 | – | – | – |
| Northeast | 98 | 1.262 | 25.437.810 | 22,1 | 26,9 | 0 | 0 | – | – | – |
| Southeast | 86 | 747 | 12.384.445 | 18,3 | 18,1 | 0 | 0 | – | – | – |
| South | 28 | 444 | 6.301.864 | 18,7 | 17,9 | 1 | 0 | – | – | – |
| Central West | 23 | 217 | 3.964.703 | 22,0 | 28,0 | 0 | 0 | – | – | – |

Clusters were defined through a *hierarchical multivariate cluster analysis* using standardized mean values of the percentage distribution of PICU bed types (I, II, and III) and regional demographic and mortality indicators. The optimal number of clusters was determined using the Bayesian Information Criterion (BIC) and the Silhouette coefficient to assess model fit and group cohesion.

¹Health Regions, defined by the Ministry of Health, with scope for high-complexity outpatient and hospital care

²Number of municipalities per Federal Unit and Health Region, according to the Brazilian Institute of Geography and Statistics (IBGE) 2022 data

³Total resident population by municipalities referenced in health regions, according to IBGE 2022 data

*Percentage of the population under 14 years, IBGE 2022 data

§Mortality rate in children under 14 per 100,000 inhabitants, according to Mortality Information System (SIM) of the Ministry of Health

#Number of institutions with PICU beds available to SUS (Unified Health System) listed in the National Register of Health Facilities (CNES)

£PICU beds in the SUS, listed in CNES

¢Percentage of PICU type I beds in CNES

¢¢Percentage of PICU type II beds in CNES

¢¢¢Percentage of PICU type III bed in CNES

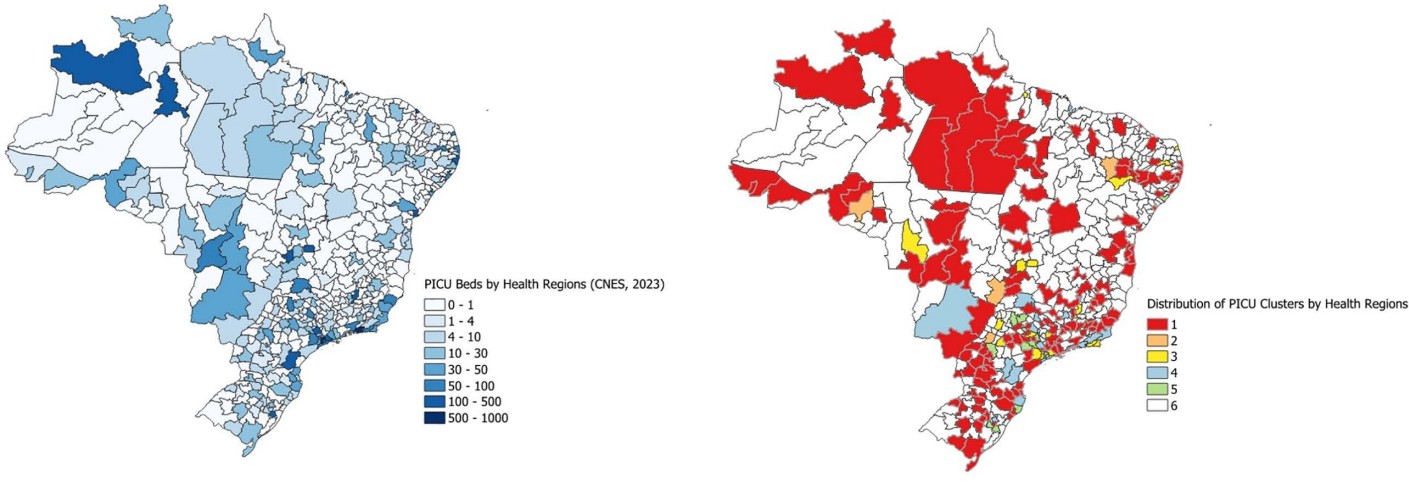

Figure 1. Geographic distribution of Pediatric Intensive Care Unit (PICU) bed availability and service clustering across Brazilian health regions (2023).
(Left) Number of PICU beds per health region based on data from the National Registry of Health Establishments (CNES), illustrating marked variation in supply.
(Right) Spatial distribution of hierarchical PICU service clusters identified through spatial analysis. Colors represent cluster levels (1–6). Together, the maps highlight substantial regional disparities in the organization, concentration, and territorial accessibility of PICU services in Brazil.

Base map derived from public-domain cartographic data provided by the Brazilian Institute of Geography and Statistics (IBGE). No copyright restrictions apply.

**Fig 1. Geographic distribution of Pediatric Intensive Care Unit (PICU) beds and service clusters across Brazilian health regions (2023). Left:** Number of PICU beds per health region according to the National Registry of Health Establishments (CNES), demonstrating substantial variation in availability. **Right:** Spatial distribution of hierarchical PICU service clusters identified through spatial analysis. Colors represent cluster levels (1–6). Together, the maps illustrate marked regional disparities in the organization, concentration, and territorial accessibility of PICU services in Brazil. *Base map derived from public-domain cartographic data provided by the Brazilian Institute of Geography and Statistics (IBGE). No copyright restrictions apply.*

One of the striking findings is that, of the approximately 6,489 PICU beds registered in the CNES database, only 50% are available to the SUS. According to official statistics, more than 80% of the Brazilian population depends exclusively on SUS services due to lack of access to private health plans or supplemental providers [19–21]. Another critical point is that 259 identified health regions—covering 2,896 municipalities and home to more than 53 million people (25% of whom are under 14 years old)—do not have any PICU beds registered in the CNES database.

To confirm and fully understand these results, audits must be conducted with healthcare providers and local regulation centers, since underreporting in CNES cannot be ruled out. If confirmed, the lack of PICU beds represents a major challenge for health managers in these regions to expand services and ensure population access to intensive pediatric care. The Ministry of Health must ultimately guarantee resource flows that support the creation or maintenance of these beds.

As previously mentioned, the presented data show significant regional differences in the distribution of PICU beds by type (I, II, III), and in child mortality rates. Regions with a higher proportion of type III beds, such as the Southeast and South, show lower pediatric mortality. Meanwhile, the North region, with fewer beds and a higher concentration of type II PICU beds, faces greater challenges in reducing child mortality.

The correlation between the number of beds and the mortality rate of children under 14 is plausible and consistent with earlier studies, both in adult critical care [22] and neonatal care [23], that emphasize the importance of health infrastructure in reducing mortality. Based on these studies, it is reasonable to assume that the availability of PICU beds with appropriate infrastructure and in sufficient quantity for the local morbidity profile can positively impact survival among critically ill children. However, it is also essential to ensure qualified staffing and continuous resource flows for complementary actions.

The cluster analysis revealed six distinct profiles based on the distribution of PICU bed types. These clusters underscore the unequal organization and availability of pediatric intensive care services in Brazil. None of the clusters reflect the national average distribution, emphasizing the need for region-specific evaluations in public policy planning.

The clusters demonstrate that the predominant types of PICU beds vary across health regions, and factors such as the concentration of more complex beds (type III) are associated with lower child mortality rates. Clusters with a predominance of lower-complexity beds (type I) or no PICU beds tend to have higher mortality rates. For example, Cluster 5—with the highest proportion of type III beds—had the lowest child mortality rate, while Cluster 6—without any PICU beds—had the highest.

This discrepancy reinforces that simply analyzing the total number of beds is insufficient. The complexity and quality of the infrastructure are key determinants of the ability to provide care and reduce child mortality. Local factors such as population distribution and technological density must also be considered, as more developed regions tend to concentrate both a greater number of beds and better outcomes—often to the detriment of peripheral or underserved areas.

The findings highlight the importance of regionalized strategies in planning the allocation of healthcare resources. Adopting approaches that consider local specificities can promote greater equity in access to pediatric intensive care beds. Additionally, the absence of a distribution pattern reflecting the national average underscores the need for decentralized, yet coordinated planning to effectively address regional healthcare demands.

Although this study suggests an association between the distribution of PICU beds and regional child mortality, these findings should be interpreted with caution. Several potential confounding factors may influence mortality rates independently of PICU availability, such as differences in socioeconomic conditions, prevalence of comorbidities, access to primary and emergency care, quality of neonatal and perinatal services, and timeliness of patient transfers. Regional variations in data reporting and health system performance may also contribute to apparent disparities. Future studies integrating these contextual variables could help clarify the complex relationship between pediatric intensive care capacity and child health outcomes.

The main limitations of this study relate to the use of secondary data from the CNES, which may not fully reflect the actual availability of PICU beds due to underreporting and inconsistencies in data entry. For example, the classification of PICU beds (Types I, II, and III) adopted in this study is based on the standards established by ANVISA Resolution RDC No. 7/2010 [4], which defines the minimum requirements for the structure, staffing, and technology of intensive care units in Brazil. These categories are consistently applied within the CNES and are used nationwide for regulatory and planning purposes. However, as CNES data depend on self-reporting by healthcare providers and periodic updates by local managers, a degree of misclassification or delayed reporting cannot be excluded. Previous evaluations have identified variability in the completeness and reliability of CNES data, particularly regarding the number of registered beds and equipment, underscoring the need for continuous validation and improvement of these systems [24,25].

Additionally, it was not possible to verify the results directly with regional regulatory centers, as access to such data requires specific authorization from local health authorities and, in some cases, ethics committee approval due to their sensitive nature.

Despite these limitations, the findings are considered robust, as they are consistent with previous studies that examined PICU availability using provider or regulatory data [26]. Moreover, this study advances beyond descriptive analyses by employing cluster analysis, which provides a nuanced understanding of regional inequalities and supports the formulation of targeted public policies to address the unequal distribution of pediatric intensive care resources in Brazil.

Finally, despite areas that could be improved, the results underscore the importance of regionalized strategies in health resource planning. Approaches that take local specificities into account may lead to more equitable access to pediatric intensive care. Moreover, the absence of a national distribution pattern highlights the necessity for decentralized—but coordinated—planning to meet regional demands effectively.

## Conclusion

The results highlight an unequal distribution of PICU beds in Brazil, with some municipalities exhibiting a desirable service supply profile, while others lack any infrastructure at all. This scenario contradicts the principles of the Unified Health

System (SUS), especially the principle of equity. Although some progress has been made, there is still a long way to go to ensure fair and efficient coverage for the entire population.

The challenge is complex and requires further studies on this topic within the context of public health. It is hoped that this study will contribute to expanding the discussion on care for critically ill children in Brazil—not only focusing on clinical care protocols but also supporting SUS management and planning efforts.

## Supporting information

**S1 Data. BD CNES DO Luis Felipe 02mar2024.**
(XLSX)

## Author contributions

**Conceptualization:** Luís Felipe Ribeiro Soares, Saint Clair Gomes Júnior, Maria Auxiliadora Souza Mendes Gomes.

**Data curation:** Luís Felipe Ribeiro Soares, Saint Clair Gomes Júnior.

**Formal analysis:** Luís Felipe Ribeiro Soares, Saint Clair Gomes Júnior, Maria Auxiliadora Souza Mendes Gomes.

**Methodology:** Luís Felipe Ribeiro Soares, Saint Clair Gomes Júnior, Maria Auxiliadora Souza Mendes Gomes.

**Supervision:** Saint Clair Gomes Júnior.

**Visualization:** Saint Clair Gomes Júnior, Maria Auxiliadora Souza Mendes Gomes.

**Writing – original draft:** Luís Felipe Ribeiro Soares, Saint Clair Gomes Júnior, Maria Auxiliadora Souza Mendes Gomes.

**Writing – review & editing:** Luís Felipe Ribeiro Soares, Saint Clair Gomes Júnior, Maria Auxiliadora Souza Mendes Gomes.

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
