## [Decision Letter · Decision Letter 0]

31 Oct 2025

Dear Dr. Soares,

Thank you for submitting your manuscript to PLOS ONE. After careful consideration, we feel that it has merit but does not fully meet PLOS ONE’s publication criteria as it currently stands. Therefore, we invite you to submit a revised version of the manuscript that addresses the points raised during the review process.

We look forward to receiving your revised manuscript.

Kind regards,

Eric Anthony Sribnick, MD, PhD, FAANS

Academic Editor

PLOS ONE

Journal Requirements:

**Additional Editor Comments:**

This is an interesting manuscript on healthcare inequity and distribution. The manuscript will be greatly improved by close attention to the reviewers' comments.

Please review the comments by the reviewers carefully, please include their entire review in your "response to reviewers," and please address their comments in a point-by-point fashion. I have carefully read your manuscript, their reviews, and I think that addressing their points will improve your study.

In the discussion, please remove the second mention of how causality cannot be addressed "Although causality cannot be confirmed..."  You are correct; however, mentioning this twice is redundant.

Best regards,

Eric Sribnick, MD, PhD

Academic Editor

PLOS One

Reviewers' comments:

Reviewer's Responses to Questions

**Comments to the Author**

1. Is the manuscript technically sound, and do the data support the conclusions?

Reviewer #1: Yes

Reviewer #2: Partly

2. Has the statistical analysis been performed appropriately and rigorously?

Reviewer #1: Yes

Reviewer #2: Yes

3. Have the authors made all data underlying the findings in their manuscript fully available?

Reviewer #1: Yes

Reviewer #2: Yes

4. Is the manuscript presented in an intelligible fashion and written in standard English?

Reviewer #1: Yes

Reviewer #2: Yes

Reviewer #1: Thank you for this interesting manuscript, which addresses an important equity issue in Brazil’s healthcare system. Its nationwide scope and robust cluster analysis strengthen its contribution to public health planning. The manuscript is technically sound. The methodology is well described and appropriate. Cluster analysis is appropriate for exploring the distribution patterns and identifying groups of regions with similar PICU availability/mortality characteristics. The results consistently support the conclusions that regional inequalities exist and that higher-complexity PICU beds are possibly associated with lower child mortality. Overall, the writing is intelligible and does not hinder comprehension. Here a few suggestions to improve the understanding and readability of the results and recommendations:

•The abstract describes the top-level results of the study, emphasizing the equity gap and possible relationship between healthcare infrastructure and infant mortality. It is well written and has “journalistic appeal”. However, for scientific purposes it should include a brief rationale, describe the study design, objective, source, volume and coverage of data, time frame, significant variables, the basis for defining clusters and be more specific on the recommendations.

•The introduction can be abridged, moving references to similar studies to the discussion section.

•Please provide a brief description of the source of data (Brazilian databases) for the benefit of unfamiliar readers.

•Consider visual aids: a map of Brazil showing cluster regions could enhance the spatial/geographical understanding of inequalities.

•Please provide details on the cluster analysis preprocessing (e.g., normalization of variables), values and metrics used to strengthen reproducibility.

•Include a footnote on the tables to explain how the clusters were defined.

•In some regions, mortality rates appear high relative to their PICU distribution. Consider adding a brief discussion of confounding factors.

•Please review language for consistency, clarity and flow. Long sentences can be split for clarity.

•Regarding limitations, the possibility of bias in the quality of data, completeness, and confounders should be discussed.

Reviewer #2: Investigators in this study map the distribution and PICU complexity level across Brazil, relating this to under-14 mortality. Results show a regional maldistribution among the country and an associated higher mortality in areas where PICU bed are less. Results revealed that though >80% of Brazil population relied on SUS , only around 50% beds were available SUS, while a large number of health regions (259) had no registered PICU bed. The regions with more Type III beds had lower under 14 mortality.

This is a very important health system data, identifying gaps and needs in PICU and correlating this with under 14 mortality.

Following points require further details/thinking.

Major Comments:

1. The paper sometimes reads as if higher Type III concentration is associated with lower mortality. This is an ecological analysis with many confounders (HDI, poverty, urbanicity, primary care coverage, transport time, NICU density, overall hospital capacity, vaccination rates, seasonal burden, case-mix). Clarify that findings are correlational, and temper causal language. Sharing the economic details / linking these with areas along with mortality data and PICU bed will be more useful before we link causality of lack of beds to mortality directly. We know economic wellbeing of an area is also linked to health outcomes. I would also urge investigators to use GIS mapping to show cluster, PICU bed, SDI level and mortality.

2. Please expand more on types of PICUs (type I, II and III. Are these based on locations too? Like type I at district level, level III at tertiary care? Are they equivalent to Level I, II and III PICU? Define more on type of basic and advanced monitoring. Because this is key information to understand the results better and advocate for more resources and also understand which kind of bed are needed where. Use some standard reference like AAP etc

3. Methods state “mortality rate of those under 14” calculated as deaths <14 / total municipal population. That yields a crude rate, not an age-specific mortality rate. To avoid compositional bias (areas with fewer children will appear to have lower “under-14 mortality”), please compute per 100,000 children <14 (deaths <14 / population <14 ×100,000) and re-run key analyses. Report both crude and age-specific rates for better understanding.

4. You reference BIC/AIC and Silhouette to choose six clusters, but the clustering model is not specified (k-means? hierarchical? model-based/Gaussian mixture? distance metric?). please clarify. Also if you used model-based clustering, BIC/AIC are appropriate—state the distributional assumptions and covariance structure. If k-means, BIC/AIC are not standard; rely on internal validity indices (Silhouette, Calinski–Harabasz, Dunn) and stability (bootstrap Jaccard).

5. You show that 259 health regions have no beds. Because CNES under-reporting is a known issue, so i would strongly urge investigators to add a sensitivity analysis: (i) alternative months (e.g., April & October 2023), (ii) cross-check with SIH-SUS ICU admissions for pediatrics, where possible, and (iii) robustness to counting temporary/surge beds.

Minor comments:

1. You have alternately cited July 2023 (Abstract/Methods–Context) and June 2023 (Eligibility, Results) as the reference month. Please reconcile (and explain the choice if July was selected for seasonal peak). Ensure all tables/figures reflect the same month and label them explicitly

2. Define the mortality period (“during the study period”): calendar year 2023? multi-year average?

3. Confirm feature scaling (z-scores). Mixing counts (beds) and proportions (Type III share) without scaling can let high-population regions dominate partitions.

4. The manuscript states >80% of the population depends on SUS, yet your analytic population is 173.1M (although Brazil’s population was ~203M in 2023). Explain the denominator (exclusion of some municipalities? aggregation by “health macro-regions”?) and ensure coverage of all municipalities/regions or justify exclusions.

5. You rely on CNES Type I/II/III labels. Provide a brief validation comment (e.g., concordance with ANVISA criteria or audits) or acknowledge misclassification risk. If feasible, sample a subset of providers to verify coding.

6. Since transfers are a key mechanism linking capacity to mortality, consider including a proxy of access (mean distance/time to nearest PICU III, or proportion of municipal population within X km of a Type III bed). Even a simple centroid-to-facility distance strengthens the access argument.

7.

Other minor language related and other comments;

1. Abstract: tighten and remove duplication; fix “adolescentes” → “adolescents.” Include the reference month/year and the unit for mortality rates.

2. Terminology: use “health regions” vs “health macro-regions” consistently; define both once. Clarify “providers” vs “units” vs “establishments.”

3. Acronyms: define CNES, DATASUS, IBGE, SIM on first use in the main text (even if in Introduction).

**Do you want your identity to be public for this peer review?** For information about this choice, including consent withdrawal, please see our Privacy Policy

Reviewer #1: **Yes: ** Fabio Zicker

Reviewer #2: No

---

## [Author Response · Author response to Decision Letter 1]

21 Nov 2025

This revised manuscript incorporates all changes requested by the Academic Editor, Dr. Eric Anthony Sribnick, and the peer reviewers. We clarified methodological limitations, emphasized that the results describe correlations rather than causation, expanded discussion of contextual and socioeconomic factors, improved figure quality and legends, and adjusted the text to strengthen transparency and alignment with PLOS ONE standards. All reviewer comments were addressed in detail in the rebuttal document, and corresponding revisions were made directly in the manuscript.

---

## [Editor Report · Decision Letter 1]

3 Dec 2025

Distribution of Pediatric Intensive Care Beds in Brazil: Regional Inequalities and Implications for Equity in the Public Health System

PONE-D-25-27587R1

Dear Dr. Soares,

We’re pleased to inform you that your manuscript has been judged scientifically suitable for publication and will be formally accepted for publication once it meets all outstanding technical requirements.

Kind regards,

Eric Anthony Sribnick, MD, PhD, FAANS

Academic Editor

PLOS ONE

Additional Editor Comments (optional):

The authors have addressed the reviewers' critiques adequately.
---

## [Editor Report · Acceptance letter]

PONE-D-25-27587R1

PLOS One

Dear Dr. Soares,

I'm pleased to inform you that your manuscript has been deemed suitable for publication in PLOS One. Congratulations! Your manuscript is now being handed over to our production team.

Kind regards,

on behalf of

Dr. Eric Anthony Sribnick

Academic Editor

PLOS One